# On Binary Classification in Extreme Regions

Hamid Jalalzai, Stephan Clémençon and Anne Sabourin

LTCI Telecom ParisTech, Université Paris-Saclay
75013, Paris, France
first.last@telecom-paristech.fr

## Abstract

In pattern recognition, a random label $Y$ is to be predicted based upon observing a random vector $X$ valued in $\mathbb{R}^d$ with $d \geq 1$ by means of a classification rule with minimum probability of error. In a wide variety of applications, ranging from finance/insurance to environmental sciences through teletraffic data analysis for instance, extreme (*i.e.* very large) observations $X$ are of crucial importance, while contributing in a negligible manner to the (empirical) error however, simply because of their rarity. As a consequence, empirical risk minimizers generally perform very poorly in extreme regions. It is the purpose of this paper to develop a general framework for classification in the extremes. Precisely, under non-parametric heavy-tail assumptions for the class distributions, we prove that a natural and asymptotic notion of risk, accounting for predictive performance in extreme regions of the input space, can be defined and show that minimizers of an empirical version of a non-asymptotic approximant of this dedicated risk, based on a fraction of the largest observations, lead to classification rules with good generalization capacity, by means of maximal deviation inequalities in low probability regions. Beyond theoretical results, numerical experiments are presented in order to illustrate the relevance of the approach developed.

## 1 Introduction

Because it covers a wide range of practical applications and its probabilistic theory can be straightforwardly extended to some extent to various other prediction problems, binary classification can be considered as the flagship problem in statistical learning. In the standard setup, $(X, Y)$ is a random pair defined on a certain probability space with (unknown) joint probability distribution $P$, where the (output) r.v. $Y$ is a binary label, taking its values in $\{-1, +1\}$ say, and $X$ models some information, valued in $\mathbb{R}^d$ and hopefully useful to predict $Y$. In this context, the goal pursued is generally to build, from a training sample $\mathcal{D}_n = \{(X_1, Y_1), \ldots, (X_n, Y_n)\}$ composed of $n \geq 1$ i.i.d. realizations of $(X, Y)$, a classifier $g : \mathbb{R}^d \to \{-1, +1\}$ minimizing the probability of error $L_P(g) = \mathbb{P}\{Y \neq g(X)\}$. The Empirical Risk Minimization paradigm (ERM in abbreviated form, see *e.g.* [5]) suggests considering solutions $g_n$ of the minimization problem $\min_{g \in \mathcal{G}} \widehat{L}_n(g)$, where $\widehat{L}_n(g)$ is a statistical estimate of the risk $L(g)$. In general the empirical version $\widehat{L}_n(g) = (1/n) \sum_{i=1}^{n} \mathbb{1}\{Y_i \neq g(X_i)\}$ is considered, denoting by $\mathbb{1}\{\mathcal{E}\}$ the indicator function of any event $\mathcal{E}$. This amounts to replacing $P$ in $L_P$ with the empirical distribution of the $(X_i, Y_i)$'s. The class $\mathcal{G}$ of predictive rules is supposed to be rich enough to contain a reasonable approximant of the minimizer of $L_P$, *i.e.* the Bayes classifier $g^*(x) = 2\, \mathbb{1}\{\eta(x) \geq 1/2\} - 1$, where $\eta(X) = \mathbb{P}\{Y = 1 \mid X\}$ denotes the posterior probability.

Because extreme observations $X$, *i.e.* observations whose norm $\|X\|$ exceeds some large threshold $t > 0$, are rare and thus underrepresented in the training dataset $\mathcal{D}_n$ classification errors in these

regions of the input space may have a negligible impact on the global prediction error of $\widehat{g}_n$. Notice incidentally that the threshold $t$ may depend on $n$, since 'large' should be naturally understood as large w.r.t the vast majority of data previously observed. Using the total probability formula, one may indeed write

$$L_P(g) = \mathbb{P}\{\|X\| > t\}\mathbb{P}\{Y \neq g(X) \mid \|X\| > t\} + \mathbb{P}\{\|X\| \leq t\}\mathbb{P}\{Y \neq g(X) \mid \|X\| \leq t\}. \quad (1)$$

Hence, due to the extremely small order of magnitude of $\mathbb{P}\{\|X\| > t\}$ and of its empirical counterpart, there is no guarantee that the standard ERM strategy produces an optimal classifier on the extreme region $\{x : \|x\| > t\}$. In other words the quantity $\mathbb{P}\{Y \neq \widehat{g}_n(X) \mid \|X\| > t\}$ may not be nearly optimal, whereas in certain practical applications (*e.g.* finance, insurance, environmental sciences, aeronautics safety), accurate prediction in extreme regions is crucial.

The purpose of the subsequent analysis is to investigate the problem of building a classifier such that the first term of the decomposition (1) is asymptotically minimum as $t \to +\infty$. We thus consider the conditional probability of error, which quantity is next referred to as the *classification risk above level* $t$, given by

$$L_t(g) := L_{P_t}(g) = \mathbb{P}\{Y \neq g(X) \mid \|X\| > t\}, \quad (2)$$

denoting by $P_t$ the conditional distribution of $(X, Y)$ given $\|X\| > t$. In this paper, we address the issue of learning a classifier $g_n$ whose risk $L_t(g_n)$ is asymptotically minimum as $t \to \infty$ with high probability. In order to develop a framework showing that a variant of the ERM principle tailored to this statistical learning problem leads to predictive rules with good generalization capacities, (non-parametric) distributional assumptions related to the tail behavior of the class distributions $F_+$ and $F_-$, the conditional distributions of the input r.v. $X$ given $Y = +/-1$, are required. Precisely, we assume that they are both multivariate regularly varying, which correspond to a large non-parametric class of (heavy-tailed) distributions, widely used in applications where the impact of extreme observations should be enhanced, or at least not neglected. Hence, under appropriate non-parametric assumptions on $F_+$ and $F_-$, as well as on the tail behavior of $\eta(x)$, we prove that $\min_g L_t(g)$ converges to a quantity denoted by $L^*_\infty$ and referred to as the *asymptotic risk in the extremes*, as $t \to \infty$. It is also shown that this limit can be interpreted as the minimum classification error related to a (non observable) random pair $(X_\infty, Y_\infty)$, whose distribution $P_\infty$ corresponds to the limit of the conditional distribution of $(X, Y)$ given $\|X\| > t$, for an appropriate normalization of $X$, as $t \to \infty$. With respect to the goal set above we next investigate the performance of minimizer $\widehat{g}_{n,\tau}$ of an empirical version of the risk $L_{P_{t_\tau}}$, where $t_\tau$ is the $(1 - \tau)$ quantile of the r.v. $\|X\|$ and $\tau \ll 1$. The computation of $\widehat{g}_{n,\tau}$ involves the $\lfloor n\tau \rfloor$ input observations with largest norm, and the minimization is performed over a collection of classifiers of finite VC dimension. Based on a variant of the VC inequality tailored to low probability regions, rate bounds for the deviation $L_t(\widehat{g}_{n,\tau}) - L^*_\infty$ are established, of order $O_\mathbb{P}(1/\sqrt{n\tau})$ namely. These theoretical results are also illustrated by preliminary experiments based on synthetic data.

The rest of the paper is organized as follows. Multivariate extreme value theory (MEVT) notions involved in the framework we develop are described in section 2, together with the probabilistic setup we consider for classification in the extremes. A notion of risk tailored to this statistical learning task is also introduced therein. Section 3 investigates how to extend the ERM principle in this situation. In particular, probability bounds proving the generalization ability of minimizers of a non-asymptotic approximant of the risk previously introduced are established. Illustrative numerical results are displayed in section 4, while several concluding remarks are collected in section 5. Some technical details and proofs are deferred to the Supplementary Material.

## 2  Probabilistic Framework - Preliminary Results

We start off with recalling concepts pertaining to MEVT and next develop a general framework in order to formulate the problem of binary classification in the extremes in a rigorous manner. For completeness, additional details about regular variation and vague convergence are given in the supplementary material (Appendix A).

### 2.1  Regularly Varying Random Vector

By definition, heavy-tail phenomena are those which are ruled by very large values, occurring with a far from negligible probability and with significant impact on the system under study. When the

phenomenon of interest is described by the distribution of a univariate random variable, the theory of regularly varying functions provides the appropriate mathematical framework for the study of heavy-tailed distributions. One may refer to [11] for an excellent account of the theory of regularly varying functions and its application to the study of heavy-tailed distributions. For examples of works where such assumptions are considered in the context of statistical learning, see *e.g.* [6, 3, 12, 10, 1] or [8]. Let $\alpha > 0$, a random variable $X$ is said to be regularly varying with tail index $\alpha$ if

$$\mathbb{P}\{X > tx \mid X > t\} \xrightarrow[t \to \infty]{} x^{-\alpha}, \qquad x > 1.$$

This is the case if and only if there exists a function $b : \mathbb{R}_+ \to \mathbb{R}_+^*$ with $b(t) \to \infty$ such that for all $x > 0$, the quantity $t\mathbb{P}\{X/b(t) > x\}$ converges to some limit $h(x)$ as $t \to \infty$. Then $b$ may be chosen as $b(t) = t^{1/\alpha}$ and $h(x) = cx^{-\alpha}$ for some $c > 0$. Based on this characterization, the heavy-tail model can be extended to the multivariate setup. Consider a $d$-dimensional random vector $X = (X^{(1)}, \ldots, X^{(d)})$ taking its values in $\mathbb{R}_+^d$. Assume that all the $X^{(j)}$ are regularly varying with index $\alpha > 0$. Then the random *vector* $X$ is said to be regularly varying with tail index $\alpha$ if there exists a non null positive Radon measure $\mu$ on the punctured space $E = [0, \infty]^d \backslash \{0\}$ and a function $b(t) \to \infty$ such that for all Borel set $A \subset E$ such that $0 \notin \partial A$ and $\mu(\partial A) = 0$,

$$t\mathbb{P}\{X/b(t) \in A\} \xrightarrow[t \to \infty]{} \mu(A).$$

In such a case, the so-called *exponent measure* $\mu$ fulfills the homogeneity property $\mu(tC) = t^{-\alpha}\mu(C)$ for all $t > 0$ and any Borel set $C \subset E$. This suggests a decomposition of $\mu$ into a radial component and an angular component $\Phi$. For all $x = (x_1, \ldots, x_d) \in \mathbb{R}_+^d$, set

$$\begin{cases} R(x) = \|x\|, \\ \Theta(x) = \left( \dfrac{x_1}{R(x)}, \ldots, \dfrac{x_d}{R(x)} \right) \in S, \end{cases}$$

where $S$ is the positive orthant of the unit sphere in $\mathbb{R}^d$ for the chosen norm $\| \cdot \|$. The choice of the norm is unimportant as all norms are equivalent in $\mathbb{R}^d$. Define an *angular measure* $\Phi$ on $S$ as

$$\Phi(B) = \mu\{r\theta \ : \ \theta \in B, r \geq 1\}, \quad B \subset S, \text{measurable}.$$

The angular measure $\Phi$ is finite, and the conditional distribution of $(R(X)/t, \Theta(X))$ given that $R(X) > t$ converges as $t \to \infty$ to a limit which admits the following product decomposition: for $r \geq 1$ and $B \subset S$ such that $\Phi(\partial B) = 0$,

$$\lim_{t \to \infty} \mathbb{P}\{R(X)/t > r, \Theta(X) \in B \mid R(X) > t\} = c\,\mu\{x \ : \ R(x) > r, \Theta(x) \in B\}$$
$$= c\,r^{-\alpha}\,\Phi(B),$$

where $c = \mu\{x \ : \ R(x) > 1\}^{-1} = \Phi(S)^{-1}$ is a normalizing constant. Thus $c\Phi$ may be viewed as the limiting distribution of $\Theta(X)$ given that $R(X)$ is large.

**Remark 1** *It is assumed above that all marginal distributions are tail equivalent to the Pareto distribution with index $\alpha$. In practice, the tails of the marginals may be different and it may be convenient to work with marginally standardized variables, that is, to separate the margins $F_j(x_j) = \mathbb{P}\{X^{(j)} \leq x_j\}$ from the dependence structure in the description of the joint distribution of $X$. Consider the standardized variables $V^{(j)} = 1/(1 - F_j(X^{(j)})) \in [1, \infty]$ and $V = (V^{(1)}, \ldots, V^{(d)})$. Replacing $X$ by $V$ permits to take $\alpha = 1$ and $b(t) = t$.*

## 2.2 Classification in the Extremes - Assumptions, Criterion and Optimal Elements

We place ourselves in the binary classification framework recalled in the introduction. For simplicity, we suppose that $X$ takes its values in the positive orthant $\mathbb{R}_+^d$. The general aim is to build from training data in the extreme region (*i.e.* data points $(X_i, Y_i)$ such that $\|X_i\| > t_n$ for a large threshold value $t_n > 0$) a classifier $g_n(x)$ with risk $L_{t_n}(g_n)$ defined in (2) being asymptotically minimum as $t_n \to \infty$. In this purpose, we introduce general assumptions guaranteeing that the minimum risk $L_t(g^*)$ above level $t$ has a limit as $t \to \infty$. Throughout the article, we assume that the class distributions $F_+$ and $F_-$ are heavy-tailed with same index $\alpha = 1$.

**Assumption 1** *For all $\sigma \in \{-, +\}$, the conditional distribution of $X$ given $Y = \sigma 1$ is regularly varying with index $1$ and angular measure $\Phi_\sigma(d\theta)$ (respectively, exponent measure $\mu_\sigma(dx)$): for $A \subset [0, \infty]^d \setminus \{0\}$ a measurable set such that $0 \notin \partial A$ and $\mu(\partial A) \neq 0$,*

$$t\mathbb{P}\left\{t^{-1}X \in A \mid Y = \sigma 1\right\} \xrightarrow[t\to\infty]{} \mu_\sigma(A), \qquad \sigma \in \{-,+\},$$

*and for $B \subset S$ a measurable set,*

$$\Phi_\sigma(B) = \mu_\sigma\{x \in \mathbb{R}_+^d : R(x) > 1, \Theta(x) \in B\}, \qquad \sigma \in \{-,+\}.$$

Under the hypothesis above, $X$'s marginal distribution, given by $F = pF_+ + (1-p)F_-$, where $p = \mathbb{P}\{Y = +1\} > 0$, is heavy-tailed as well with index $1$. Indeed, we have:

$$t\mathbb{P}\left\{t^{-1}X \in A\right\} \xrightarrow[t\to\infty]{} \mu(A) := p\mu_+(A) + (1-p)\mu_-(A).$$

And similarly

$$\Phi(B) := p\Phi_+(B) + (1-p)\Phi_-(B).$$

Observe also that the limiting class balance can be expressed using the latter asymptotic measures. Indeed, let $\Omega = \{x \in \mathbb{R}_+^d : \|x\| \leq 1\}$ denote the positive orthant of the unit ball and let $\Omega^c$ denote its complementary set in $\mathbb{R}_+^d$. We have:

$$p_t = \mathbb{P}\{Y = +1 \mid \|X\| > t\} = \frac{t\mathbb{P}\{\|X\| > t \mid Y = 1\}p}{t\mathbb{P}\{\|X\| > t\}} \xrightarrow[t\to\infty]{} p\frac{\mu_+(\Omega^c)}{\mu(\Omega^c)} = p\frac{\Phi_+(S)}{\Phi(S)} \tag{3}$$
$$\stackrel{\text{def}}{=} p_\infty.$$

**Remark 2** (ON ASSUMPTION 1) *We point out that only the situation where the supposedly heavy-tailed class distributions $F_+$ and $F_-$ have the same tail index is of interest. Suppose for instance that the tail index $\alpha_+$ of $F_+$ is strictly larger than that of $F_-$, $\alpha_-$, that is $F_-$ has heavier tail than $F_+$. In such a case $F$ is still regularly varying with index $\min\{\alpha_+, \alpha_-\}$ and $p_t \to 0$. In this case, one may straightforwardly see that the classifier predicting always $-1$ on $\{x \in \mathbb{R}_+^d : \|x\| > t\}$ is optimal as $t$ increases to infinity.*

**Remark 3** (ON ASSUMPTION 1 (BIS)) *As noticed in Remark 1, assuming that $\alpha = 1$ is not restrictive when the marginal distributions are known. In practice however, they must be estimated. Due to space limitations, in the present analysis, we shall neglect the estimation error arising from their estimation. Relaxing this assumption, as made in e.g. [7], will be the subject of future work.*

**Asymptotic criterion for classification in the extremes.** The goal pursued is to construct a classifier $g_n$, based on the training examples $\mathcal{D}_n$, minimizing the *asymptotic risk in the extremes* given by

$$L_\infty(g) = \limsup_{t\to\infty} L_t(g). \tag{4}$$

We also set $L_\infty^* = \inf_{g \text{ measurable}} L_\infty(g)$. It is immediate that any classifier which coincides with the Bayes classifier $g^*$ on the region $t\Omega^c = \{x \in \mathbb{R}_+^d : \|x\| > t\}$ is optimal w.r.t. the distribution $P_t$. In particular $g^*$ minimizes $L_t$ and the associated risk is

$$L_t^* := L_t(g^*) = \mathbb{E}\left[\min\{\eta(X), 1 - \eta(X)\} \mid \|X\| > t\right], \qquad t > 0. \tag{5}$$

Thus, for all classifier $g, L_t(g) \geq L_t(g^*)$, and taking the limit superior shows that $g^*$ minimizes $L_\infty$, that is $L_\infty^* = L_\infty(g^*)$.

**Optimality.** The objective formulated above can be connected with a standard binary classification problem, related to a random pair $(X_\infty, Y_\infty)$ taking its values in the limit space $\Omega^c \times \{-1, +1\}$, see Theorem 1 below. Let $\mathbb{P}\{Y_\infty = +1\} = p_\infty$ as in (3) and define the distribution of $X_\infty$ given that $Y_\infty = \sigma 1$, $\sigma \in \{-, +\}$ as $\mu_\sigma(\Omega^c)^{-1}\mu_\sigma(\cdot)$. Then for $A \subset \Omega^c$, using (3),

$$\begin{aligned}
\mathbb{P}\{X_\infty \in A, Y_\infty = +1\} &= \frac{p_\infty\mu_+(A)}{\mu_+(\Omega^c)} = \frac{p\mu_+(A)}{\mu(\Omega^c)} = \frac{p\lim_t t\mathbb{P}\{X \in tA \mid Y = +1\}}{\lim_t t\mathbb{P}\{X \in t\Omega^c\}} \\
&= \lim_{t\to\infty} \mathbb{P}\{X \in tA, Y = +1 \mid \|X\| > t\}.
\end{aligned}$$

We denote by $P_\infty$ the joint distribution of $(X_\infty, Y_\infty)$ thus defined. As shall be seen below, under appropriate and natural assumptions, classifiers with minimum asymptotic risk in the extremes are in 1-to-1 correspondence with solutions of the binary classification problem related to $(X_\infty, Y_\infty)$. Let $\rho$ be a common dominating measure for $\Phi_-, \Phi_+$ on $S$ ($\rho$ does not need to be the Lebesgue measure, take *e.g.* $\rho = \Phi_+ + \Phi_-$). Then denote by $\varphi_+, \varphi_-$ respectively the densities of $\Phi_+, \Phi_-$ w.r.t. $\rho$. By homogeneity of $\mu_+, \mu_-$, the conditional distribution of $Y_\infty$ given $X_\infty = x$ is

$$
\eta_\infty(x) \stackrel{\text{def}}{=} \mathbb{P}\{Y_\infty = 1 \mid X_\infty = x\} = \frac{p_\infty \varphi_+(\Theta(x))/\Phi_+(S)}{p_\infty \varphi_+(\Theta(x))/\Phi_+(S) + (1-p_\infty)\varphi_-(\Theta(x))/\Phi_-(S)}
$$

$$
= \frac{p\varphi_+(\Theta(x))}{p\varphi_+(\Theta(x)) + (1-p)\varphi_-(\Theta(x))}.
$$

Notice that $\eta_\infty$ is independent of the chosen reference measure $\rho$ and that $\eta_\infty$ is constant along rays, that is $\eta_\infty(tx) = \eta_\infty(x)$ for $(t, x)$ such that $\min(\|tx\|, \|x\|) \geq 1$. The optimal classifier for the random pair $(X_\infty, Y_\infty)$ with respect to the classical risk $L_{P_\infty}$ is clearly

$$
g_\infty^*(x) = 2\mathbb{1}\{\eta_\infty(x) \geq 1/2\} - 1.
$$

Again $g_\infty^*$ is constant along rays on $\Omega^c$ and is thus a function of $\Theta(x)$ only. We abusively denote $\eta_\infty(x) = \eta_\infty(\Theta(x))$. The minimum classification error is

$$
L_{P_\infty}^* = L_{P_\infty}(g_\infty^*) = \mathbb{E}\left[\min\{\eta_\infty(\Theta_\infty), 1 - \eta_\infty(\Theta_\infty)\}\right], \tag{6}
$$

where $\Theta_\infty = \Theta(X_\infty)$. More generally, observe that any class $\mathcal{G}_S$ of classifiers $g : \theta \in S \mapsto g(\theta) \in \{-1, +1\}$ defines a class of classifiers on $\mathbb{R}_+^d$, $x \in \mathbb{R}_+^d \mapsto g(\Theta(x))$, that shall still be denoted by $\mathcal{G}_S$ for simplicity. The next result claims that, under the regularity hypothesis stated below, the classifier $g_\infty^*$ is optimal for the asymptotic risk in the extremes, that is $L_\infty(g_\infty^*) = \inf_g L_\infty(g)$. We shall also prove that $L_\infty(g_\infty^*) = L_{P_\infty}^*$.

**Assumption 2** (UNIFORM CONVERGENCE ON THE SPHERE OF $\eta(tx)$) *The limiting regression function $\eta_\infty$ is continuous on $S$ and*

$$
\sup_{\theta \in S} |\eta(\Theta(t\theta)) - \eta_\infty(\theta)| \xrightarrow[t \to \infty]{} 0
$$

**Remark 4** (ON ASSUMPTION 2) *By invariance of $\eta_\infty$ along rays, Assumption 2 is equivalent to*

$$
\sup_{\{x \in \mathbb{R}_+^d : \|x\| \geq t\}} |\eta(x) - \eta_\infty(x)| \xrightarrow[t \to \infty]{} 0.
$$

*Assumption 2 is satisfied whenever the probability densities $f_+, f_-$ of $F_+, F_-$ are continuous, regularly varying with limit functions $q_+, q_-$, and when the convergence is uniform, that is if*

$$
\lim_{t \to \infty} \sup_{x \in S} |t^{d+1} f_\sigma(tx) - q_\sigma(x)| = 0, \qquad \sigma \in \{+, -\}. \tag{7}
$$

*In such a case $q_+, q_-$ are respectively the densities of $\mu_+, \mu_-$ with respect to the Lebesgue measure and are continuous, which implies the continuity of $\varphi_+, \varphi_-$. The latter uniform convergence assumption is introduced in [4] and is used* e.g. *in [2] in the context of minimum level sets estimation.*

**Theorem 1** (OPTIMAL CLASSIFIERS IN THE EXTREMES) *Under Assumptions 1 and 2,*

$$
L_t^* \xrightarrow[t \to \infty]{} L_{P_\infty}^*. \tag{8}
$$

*Hence, we have: $L_\infty^* = L_{P_\infty}^*$. In addition, the classifier $g_\infty^*$ minimizes the asymptotic risk in the extremes:*

$$
\inf_{g \text{ measurable}} L_\infty(g) = L_\infty(g_\infty^*) = \mathbb{E}\{\min(\eta_\infty(\Theta_\infty), 1 - \eta_\infty(\Theta_\infty))\}.
$$

Refer to the Supplementary Material for the technical proof. Theorem 1 gives us the form of the optimal classifier in the extremes $g_\infty^*(x) = g_\infty^*(\Theta(x))$, which depends only on the angular component $\Theta(x)$, not the norm $R(x)$. This naturally leads to applying the ERM principle to a collection of classifiers of the form $g(x) = g(\Theta(x))$ on the domain $\{x \in \mathbb{R}_+^d : \|x\| > t\}$ for $t > 0$ large enough. The next section provides statistical guarantees for this approach.

# 3 Empirical Risk Minimization in the Extremes

Consider a class $\mathcal{G}_S$ of classifiers $g : \theta \in S \mapsto g(\theta) \in \{-1, +1\}$ on the sphere $S$. It also defines a collection of classifiers on $\mathbb{R}_+^d$, namely $\{g(\Theta(x)) : g \in \mathcal{G}_S\}$, which we denote by $\mathcal{G}_S$ for simplicity. Sorting the training observations by decreasing order of magnitude, we introduce the order statistics $\|X_{(1)}\| > \ldots > \|X_{(n)}\|$ and we denote by $Y_{(i)}$ the corresponding sorted labels. Fix a small fraction $\tau > 0$ of extreme observations, and let $t_\tau$ be the quantile at level $(1 - \tau)$ of the r.v. $\|X\|$: $\mathbb{P}\{\|X\| > t_\tau\} = \tau$. Set $k = \lfloor n\tau \rfloor$ and consider the empirical risk

$$\widehat{L}_k(g) = \frac{1}{k} \sum_{i=1}^{k} \mathbf{1}\{Y_{(i)} \neq g(\Theta(X_{(i)}))\} = L_{\widehat{P}_k}(g),  \tag{9}$$

where $\widehat{P}_k$ denotes the empirical distribution of the truncated training sample $\{(X_i, Y_i) : \|X_i\| \geq \|X_k\|, \ i \in \{1, \ldots, n\}\}$, the statistical version of the conditional distribution $P_{t_\tau}$. We now investigate the performance in terms of asymptotic risk in the extremes $L_\infty$ of the solutions of the minimization problem

$$\min_{g \in \mathcal{G}_S} \widehat{L}_k(g).  \tag{10}$$

The practical issue of designing efficient algorithms for solving (10) is beyond the scope of this paper. Focus is here on the study of the learning principle that consists in assigning to any very large input value $x$ the likeliest label based on the direction $\Theta(x)$ it defines only (the construction is summarized in Algorithm 1 below). The following result provides an upper bound for the excess of classification error in the domain $t_\tau \Omega^c$ of solutions of (10). Its proof, which relies on a maximal deviation inequality tailored to low probability regions, is given in the Supplementary Material.

**Theorem 2** *Suppose that the class $\mathcal{G}_S$ is of finite VC dimension $V_{\mathcal{G}_S} < +\infty$. Let $\widehat{g}_k$ be any solution of* (10). *Recall $k = \lfloor n\tau \rfloor$. Then, for $\delta \in (0, 1)$, $\forall n \geq 1$, we have with probability larger than $1 - \delta$:*

$$L_{t_\tau}(\widehat{g}_k) - L_{t_\tau}^* \leq \frac{1}{\sqrt{k}} \left( \sqrt{2(1-\tau)\log(2/\delta)} + C\sqrt{V_{\mathcal{G}_S} \log(1/\delta)} \right)$$
$$+ \frac{1}{k} \left( 5 + 2\log(1/\delta) + \sqrt{\log(1/\delta)}(C\sqrt{V_{\mathcal{G}_S}} + \sqrt{2}) \right) + \left\{ \inf_{g \in \mathcal{G}_S} L_{t_\tau}(g) - L_{t_\tau}^* \right\},$$

*where $C$ is a constant independent from $n$, $\tau$ and $\delta$.*

**Remark 5** (ON MODEL SELECTION) *Selecting an appropriate model class $\mathcal{G}_S$ is a crucial issue in machine-learning. Following in the footsteps of structured risk minimization, one may use a VC bound for $\mathbb{E}[\sup_{g \in \mathcal{G}_S} |\widehat{L}_k(g) - \mathbb{E}[\widehat{L}_k(g)]|]$ as a complexity regularization term to penalize in an additive fashion the empirical risk* (9). *Considering a collection of such models, oracle inequalities guaranteeing the quasi-optimality of the rule minimizing the penalized empirical risk can be then classically established by means of a slight modification of the argument of Theorem 2's proof, see e.g. Chapter 18 in [5].*

The upper bound stated above shows that the learning rate is of order $O_\mathbb{P}(1/\sqrt{k})$, where $k$ is the actual size of the training data set used to perform approximate empirical risk minimization in the extremes. As revealed by the corollary below, this strategy permits to build a consistent sequence of classifiers for the $L_\infty$-risk, when the fraction $\tau = \tau_n$ decays at an appropriate rate (provided that the model bias can be neglected of course).

**Corollary 1** *Suppose that the assumptions of Theorems 1-2 are fulfilled. In addition, assume that the model bias asymptotically vanishes as $\tau \to 0$, i.e.*

$$\inf_{g \in \mathcal{G}_S} L_{t_\tau}(g) - L_{t_\tau}^* \longrightarrow 0 \quad \text{as } \tau \to 0.$$

*Then, as soon as $k \to +\infty$ as $n \to \infty$, the sequence of classifiers $(\widehat{g}_k)$ is consistent in the extremes, meaning that we have the convergence in probability:*

$$L_\infty(\widehat{g}_k) \to L_\infty^* \text{ as } n \to \infty.$$

**Algorithm 1 (ERM in the extremes)**

    **Input** *Training dataset* $\mathcal{D}_n = \{(X_1, Y_1), \ldots, (X_n, Y_n)\}$, *collection* $\mathcal{G}_S$ *of classifiers on the sphere, size* $k \leq n$ *of the training set composed of extreme observations*

    *1* **Standardization.** *Standardize the input vector by applying the rank-transformation:* $\forall i \in \{1, \ldots, n\}$, $\hat{V}_i = \hat{T}(X_i)$, *where*

$$\hat{T}(x) = \left( 1 / \left( 1 - \hat{F}_j(x_j) \right) \right)_{j=1, \ldots, d},$$

    *for all* $x = (x_1, \ldots, x_d) \in \mathbb{R}^d$.

    *2* **Truncation.** *Sort the training input observations by decreasing order of magnitude*

$$\|\hat{V}_{(1)}\| \geq \ldots \geq \|\hat{V}_{(n)}\|,$$

    *and consider the set of extreme training points*

$$\left\{ (\hat{V}_{(1)}, Y_{(1)}), \ldots, (\hat{V}_{(k)}, Y_{(k)}) \right\}.$$

    *3* **Optimization.** *Compute a solution* $\widehat{g}_k(\theta)$ *of the minimization problem*

$$\min_{g \in \mathcal{G}_S} \frac{1}{k} \sum_{i=1}^{k} \mathbf{1} \left\{ Y_{(i)} \neq g \left( \Theta(\hat{V}_{(i)}) \right) \right\}$$

    **Output** *The classifier* $\widehat{g}_k \left( \Theta \left( \hat{T}(x) \right) \right)$, *applicable on the region* $\{x : \|\hat{T}(x)\| > \|\hat{V}_{(k)}\|\}$.

**Remark 6 (Choice of $k$)** *Determining the best value of $k$ is a typical challenge of Extreme Value analysis. This is typically a bias/variance trade-off, too large values introduce a bias by taking into account observations which are not large enough, so that their distribution deviates significantly from the limit distribution of extremes. On the other hand, too small values obviously increase the variance of the classifier. See* e.g.*[6] or[7] and the reference therein for a discussion. In practice a possible default choice is $k = \sqrt{n}$, otherwise cross-validation can be performed.*

## 4   Illustrative Numerical Experiments

The purpose of our experiments is to provide insights into the performance of the classifier $\widehat{g}_k$ on extreme regions constructed *via* Algorithm 1.The training set is ordered as in Step 1 of Algorithm 1. For a chosen $k$, let $t = \|\hat{T}(X_{(k)}^{\text{train}})\|$, the $L_1$ norm is used throughout our experiments. The extreme test set $\mathcal{T}$ is the subset of test points such that $\|\hat{T}(X_i^{\text{test}})\| > t$. To approximate of the asymptotic risk in the extremes $L_\infty(\widehat{g}_k)$ and illustrate the generalization ability of the proposed classifier in the extreme region, we consider decreasing subsets of $\mathcal{T}$. Namely denoting $n_{\text{test}} = |\mathcal{T}|$, we keep only the $\lfloor \kappa n_{\text{test}} \rfloor$ largest instances of $\mathcal{T}$ in terms on $\|\hat{T}(X_i^{\text{test}})\|$, for decreasing values of $\kappa \in (0, 1]$. This experimental framework is summarized in Figure 1, where $\lambda t = \|\hat{T}(X_{(\lfloor \kappa n_{\text{test}} \rfloor)}^{\text{test}})\| \geq t$.

We consider two different classification algorithms for Step 3 in Algorithm 1, namely random forest (RF) and k-nearest neighbors (k-NN), which correspond to two different classes $\mathcal{G}_S$ of classifiers. For each class $\mathcal{G}_S$, the performance of $\widehat{g}_k$ (which considers only the direction $\Theta(\hat{T}(x))$ of both training and testing data, in other words classifies the projected datasets onto the unit sphere (see Figure 2 ) is compared with that of the classical version of the algorithm (RF or k-NN) taking as input the same training data but without the standardization and truncation steps neither the projection onto the unit sphere. Figures 4 and 5 summarize the results obtained using RF respectively with a multivariate simulated dataset and with a real world dataset. The simulated dataset is generated from a logistic distribution as described in [13]. The positive and negative instances are generated using

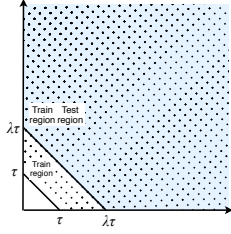
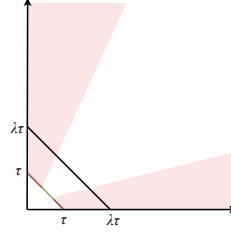

Figure 1: Train set (dotted area) and test set (colored area).

Figure 2: Colored cones correspond to a given label from the classifier on the simplex.

two different dependency parameters. An example of dataset thus obtained is displayed in Figure 3. We report the results obtained with $5 \cdot 10^3$ points for each label for the train set and $5 \cdot 10^4$ points for each label for the test set. $k = 100$ and $\kappa \in [1, 0.3]$. the number of trees for both random forests (in the regular setting and in the setting of Algorithm 1) is set to 200. The number of neighbors for both k-NN's is set to 5.

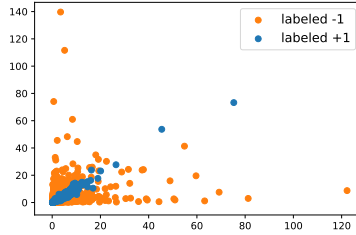

Figure 3: Toy dataset generated from a multivariate logistic distribution projected on $\mathbb{R}^2$.

The real dataset known as Ecoli dataset, introduced in [9], deals with protein localization and contains 336 instances and 8 features. The Supplementary Material gathers additional details concerning the datasets and the tuning of RF and k-NN in our experiments, as well as additional results obtained with the above described datasets and with a simulated dataset from a different distribution.

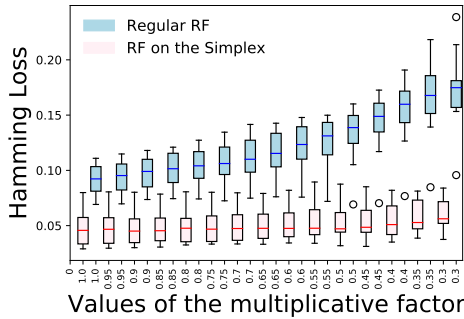

Figure 4: Logistic data - test loss of RF on the simplex and regular RF depending on the multiplicative factor $\kappa$.

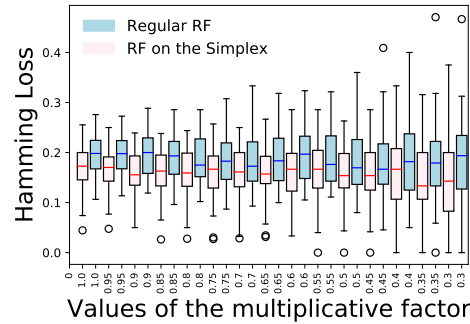

Figure 5: Real data - test loss of RF on the simplex and regular RF depending on the multiplicative factor $\kappa$.

Figure 4 shows the evolutions of the Hamming losses with decreasing values of $\kappa \in [0.3, 1]$. The boxplots display the losses obtained with 10 independently simulated datasets. For the experiment on the Ecoli dataset (Figure 5), one third of the dataset is used as a test set and the rest corresponds to the train set. $k = 100$ and $\kappa \in [0.3, 1]$ (considering smaller values of $\kappa$ was prevented by data scarcity). The boxplots display the results for different (random) partitions of the data into a train and a test set. In both examples, the loss of the regular classifier is worse (and even increases) when $\kappa$ decreases whereas the classifier resulting from the proposed approach is better and has a better extrapolation ability.

# 5 Conclusion

In various applications (*e.g.* safety/security, finance, insurance, environmental sciences), it is of prime importance to predict the response $Y$ of a system when it is impacted by shocks, corresponding to extremely large input values $X$. In this paper, we have developed a rigorous probabilistic framework for binary classification in extreme regions, relying on the (nonparametric) theory of regularly varying random vectors, and proved the accuracy of the ERM approach in this context, when the risk functional is computed from extreme observations only. The present contribution may open a new line of research, insofar as progress can be naturally expected in the design of algorithmic learning methods tailored to extreme points (or their projection onto the unit sphere) and statistical issues such as estimation of the minimum risk in the extremes, $L_\infty^*$, remain to be addressed.

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
