[Reviews · NeurIPS 2018]

Reviewer 1



This paper proposes a formalism for understanding and guaranteeing generalization in extreme regions of feature space. On the applied side, it is a very welcome and timely contribution, as it touches upon the safety and robustness of learning. On the methodological side, machine learning is bound to benefit from the years of experience of probabilists in extreme value theory. I generally like the paper quite a bit. I only have minor comments. - The paper omits any discussion of slowly varying functions, which is mostly okay, but worth mentioning (say at L92). - Worth pointing out after L130 that the \Phi also have the same additive form. - My most specific criticism is the following. In light of Remark 2, one sees that we need to assume the +/- case \alpha to be the same. But it is less clear whether we can have the asymptotic scaling (b) chosen to be the same. In particular, the standardization of the marginal of X (as in Algorithm 1) will not necessarily standardize the conditionals given Y=+/-1. Is there no repercussion to this? It seems to me that we are implicitly assuming equivalent scales. Please clarify. - L170, for clarity, use \theta instead of x. - L172, replace G, H with F+, F-. - I think on L231 you mean the norm of \hat T (of X^{train}_{(k)}). Otherwise it wouldn't make sense to switch to the standardized version on the test set, would it? Finally, this line of work would not be complete without mentioning some earlier work that is concerned with the same issue in spirit, even if the framework is slightly different (more discrete) by Roos, Grunwald, Myllymaki, and Tirri. "Generalization to unseen cases" (NIPS 2005). Even that case could benefit from extreme value theory, as is treated for example by Ohannessian and Dahleh. "Rare probability estimation under regularly varying heavy tails" (COLT 2012). [After author feedback] Thank you for your clarifications. I hope you include them in the paper, especially those pertaining to the common scaling assumption. I keep my assessment unchanged.

Reviewer 2



Summary This paper introduces theoretical analysis of learning a classifier in an extreme region, where the norm of input vector is very large. The authors describe a new definition of natural and asymptotic notion of risk for predictive performance in extreme regions. The theoretical properties of definition are fully investigated and empirical performance was validated by synthetic datasets. Strengths [Motivation] Applicable well to clinical data analysis. Sometimes, large feature values are very rarely observed, but its meaning is significantly important. [Theoretical analysis] The authors presented some interesting theoretical results on learning a classifier in extreme regions, though it’s difficult to follow every step in the proof. Weaknesses [Positive orthant assumption] I’m not quite convinced to analyze a classifier by restricting an input space to the positive orthant. We could normalize the training data to fit them in the positive orthant, but it’s rarely conducted. Moreover, in section 3, a classifier is defined on the unit sphere. All theories in section 2 could be generalized to the unit sphere? [Fixing alpha] It seems to be too restrictive to fix alpha=1. It could be possible to generalize the theory to smaller alpha values? [Lack of comparison] There exist many similar works to learn a classifier with heavy-tail distributions. This paper doesn’t compare any previous works and discussion on the difference between the proposed algorithm and existing ones. Moreover, the proposed algorithm is just learning a classifier by sub-sampling instances with large values. [Lack of real-world experiments] Since this paper doesn’t contain any results on real-world datasets, it’s very difficult to understand a practical importance of the proposed method. == Updated review == I’ve read authors’ response on all reviewers’ comments. My original concerns are fully addressed, though I still believe that empirical justification is relatively weaker than the novelty of theoretical analysis. During discussion, I’ve changed my original score to 6 (weak accept).

Reviewer 3



This article deals with the computation of dichotomies when the focus is on generalization performance in extreme regions (regions where the norm of the descriptions is large). The authors endow this topic with a theoretical framework. They introduce a variant of the Empirical Risk Minimization (ERM) inductive principle designed to select functions optimal in the extremes. An upper bound on their risks in the extremes is established for the case of function classes with finite VC dimension. This gives birth to a meta-algorithm of function selection (Algorithm 1). This algorithm is then applied to two classification methods: the random forest and the k-nearest neighbors. Two data sets are used: a synthetic one and a real-world one. Although limited, the experimental results appear promising. The authors motivate clearly their contribution. The paper is well written and seems technically sound. A weakness is the choice of the VC dimension as capacity measure. With this restriction in mind, the choice of the two classifiers involved in the illustrative experiments is strange, since they can be seen as margin classifiers. A significant improvement would be to provide the reader with hints on the difficulties raised by the multi-class extension.